# Precise localization and dynamic distribution of Japanese encephalitis virus in the rain nuclei of infected mice

**Wei Han[1], Mingxing Gao[1], Changqing Xie[1], Jinhua Zhang[1], Zikai Zhao[1], Xueying Hu[1], Wanpo Zhang[1], Xiaoli Liu[1], Shengbo Cao[1,2], Guofu Cheng[1], Changqin Gu[1]***

**1** College of Veterinary Medicine, Huazhong Agricultural University, Wuhan, Hubei, P. R. China, **2** State Key Laboratory of Agricultural Microbiology, Huazhong Agricultural University, Wuhan, Hubei, P. R. China

* guchangqin@mail.hzau.edu.cn

## Abstract

Japanese encephalitis virus (JEV) is a pathogen that causes severe vector-borne zoonotic diseases, thereby posing a serious threat to human health. Although JEV is potentially neurotropic, its pathogenesis and distribution in the host have not been fully elucidated. In this study, an infected mouse model was established using a highly virulent P3 strain of JEV. Immunohistochemistry and *in situ* hybridization, combined with anatomical imaging of the mouse brain, were used to dynamically localize the virus and construct three-dimensional (3D) images. Consequently, onset of mild clinical signs occurred in some mice at 3.5 d post JEV infection, while most mice displayed typical neurological signs at 6 d post-infection (dpi). Moreover, brain pathology revealed typical changes associated with non-suppurative encephalitis, which lasted up to 8 d. The earliest detection of viral antigen was achieved at 3 dpi in the thalamus and medulla oblongata. At 6 dpi, the positive viral antigen signals were mainly distributed in the cerebral cortex, olfactory area, basal ganglia, thalamus, and brainstem regions in mice. At 8 dpi, the antigen signals gradually decreased, and the localization of JEV tended to concentrate in the cerebrum and thalamus, while no viral antigen was detected in the brain at 21 dpi. In this model, the viral antigen was first expressed in the reticular thalamic nucleus (Rt), and the virus content is relatively stable. The expression of the viral antigen in the hippocampal CA2 region, the anterior olfactory nucleus, and the deep mesencephalic nucleus was high and persistent. The 3D images showed that viral signals were mostly concentrated in the parietal cortex, occipital lobe, and hippocampus, near the mid-sagittal plane. In the early stages of infection in mice, a large number of viral antigens were detected in denatured and necrotic neurons, suggesting that JEV directly causes neuronal damage. From the time of its entry, JEV is widely distributed in the central nervous system thereby causing extensive damage.

## Author summary

There are many theories regarding the mechanism of entry of the Japanese encephalitis virus (JEV) into the nervous system. The inflammation cascade effect, resulting from the

---

**Data Availability Statement:** All relevant data are within the manuscript and its Supporting Information files.

**Funding:** This work was supported by the National Key Research and Development Program of China (2016YFD0500407) and the National Natural Sciences Foundation of China (32030107) for CG and SC. The funders had no role in study design, data collection and analysis, decision to publish, or preparation of the manuscript.

**Competing interests:** The authors have declared that no competing interests exist.

virus entering the central nervous system (CNS), is a major cause of brain injury in JEV patients. In this study, we found that the earliest point at which viral antigen was detected in the brain tissues following peripheral infection of JEV was at 3d. The virus was located in the nerve nuclei of the thalamus and medulla oblongata and, subsequently, viral antigens were found in the anterior olfactory nucleus. At 4 dpi, the virus was extensively distributed in the brain tissue, and at 6 d -8 d the viral antigen was widely distributed and highly concentrated. The viral concentration detected in the ventromedial thalamic nucleus (VM), deep mesencephalic nucleus (DpMe), and motor trigeminal nucleus (Mo5) remained high throughout the experiment. The hypertrophic nerve nuclei of JEV include the early anterior olfactory (AO) nucleus and the late hippocampal CA2 region. In the early stages of viral infection (6 dpi), the changes in viral antigen concentration and mortality rate were consistent. It was hypothesized that this stage represents the activation of viral proliferation and brain inflammation.

## Introduction

Japanese encephalitis (JE) is a mosquito-borne zoonotic infectious disease caused by Japanese encephalitis virus (JEV), which presents a high risk in Asia and parts of the Western and South Pacific Ocean. Approximately 68,000 cases of JE are reported in Asia every year [1], posing a serious threat to public health and safety. In patients with JE, 25–30% of cases are fatal, and 50% result in permanent neurological sequelae, such as recurrent epilepsy, paralysis and intellectual disability [2].

During natural infection, the skin is the initial site for JEV entry. Keratinocytes, Langerhans cells, fibroblasts, mast cells, and monocytes/macrophages are all predisposed to JEV infection [3]. Following skin infection, Langerhans cells carrying JEV migrate to the lymph nodes [4]. The virus may enter the CNS through blood circulation or peripheral nervous system [5]. After JEV enters the CNS, it localizes in specific brain regions and causes neuronal damage [6,7]. These regions include the cerebral cortex, olfactory area, basal ganglia, hippocampus, and brainstem [8].

Autopsy studies have described classic pathological changes in the brain following JEV infection [9]. However, there is a lack of dynamic and continuous observation of these characteristic changes. Mice infected with JEV display similar clinical signs and pathological characteristics to humans [10], therefore, they are an ideal animal model for the study of JEV. The precise localization of JEV in the CNS of mice at different times post-infection has not been reported. In this study, the peripheral infection pathway was simulated through intraperitoneal injection of JEV, to enable observation of the resultant pathological changes in different brain regions, and determine the dynamic distribution of JEV in the mouse brain. This is the first study to dynamically and accurately locate JEV in the nerve nuclei.

## Results

### Clinical signs and anatomical changes in JEV infected mice

There were no apparent clinical signs in mice within 3 d post JEV infection. At 3.5 dpi, a few mice displayed ruffled fur, curling behavior, and a loss of appetite. At 4 dpi to 6 dpi, a majority of the mice showed neurological signs, characterized by an arched back, tremor, lethargy, gathering, hind limb paralysis, frequent blinking, and circling. In addition, some mice exhibited orbital hemorrhage, and 56 mice euthanized during this period. The mice recovered gradually

 

between 6 dpi and 15 dpi, such that, only a few mice showed the above neurological signs. Only 12 mice died during this period (Table 1). No abnormal clinical signs were observed 15 dpi. Edema, congestion, and hemorrhage were observed in the dissected brain tissue. The pathological changes gradually worsened between 1 dpi and 6 dpi, however, these were alleviated after 6 d. No pathological changes were apparent at 21 dpi.

## Histopathological changes in the brains of JEV infected mice

At 3 d post JEV infection, no significant histopathological changes were observed in the brain.

At 4 d post JEV infection, significant pathological changes were observed in the cerebrum, diencephalon and brainstem of the mice. In the cerebrum, the most obvious lesions occurred in the cerebral cortex, characterized by the proliferation of glial cells, infiltration of a large number of lymphocytes and segmented granulocytes around the blood vessels, and perivascular cuffing (Fig 1A). The main pathological change recorded in the brainstem was that of glial cell proliferation. Perivascular cuffing could be observed in the substantia nigra (SN) of the diencephalon and mesencephalon.

The most serious lesions were found between 5 d and 6 d post JEV infection. In the cerebral cortex, proliferation of a large number of glial cells, degeneration and necrosis of neurons, concentration and fragmentation of neuronal nuclei, enhancement of cytoplasmic eosinophilia (Fig 1B), and perivascular cuffing (Fig 1C) were observed. The predominant pathological changes in the diencephalon included mainly neuronal necrosis and glial cell proliferation, with lymphocyte infiltration around the blood vessels. Extensive neuronal necrosis and glial cell proliferation in the mesencephalon was noted, but there was no significant inflammatory reaction around the blood vessels (Fig 1D). The pathological changes in the medulla oblongata and pons mainly comprised glial cell proliferation and neurophagy (Fig 1E).

At 15 d post JEV infection, the pathological changes in the brain tissue were gradually alleviated. Glial cell proliferation and neuronal necrosis could still be observed in the cerebrum, diencephalon, and mesencephalon. In the medulla oblongata and pons, glial cell proliferation remained the primary manifestation. At 21 d post JEV infection, the brain structures of mice were essentially normal, and no significant pathological changes were observed. Furthermore, no significant histopathological changes were observed in the cerebellum during the entire infection process, and no significant pathological changes were recorded in the brain tissue of the control group (Fig 1F).

## Localization of JEV by immunohistochemistry and *in situ* hybridization

After consecutive sectioning of the brain tissue, JEV was localized by immunohistochemistry (IHC) and *in situ* hybridization (ISH) staining. The IHC results revealed that positive signal for the virus antigen was distributed in the cytoplasm of the whole neuron. The JEV signal was detected in the cerebrum (Fig 2A), mesencephalon, pons, medulla oblongata (Fig 2B), and

**Table 1. The death of mice infected with Japanese encephalitis virus P3.**

| Time after infection (d) | Number of deaths |
|---|---|
| 0–4 | 0 |
| 4–5 | 28 |
| 5–6 | 28 |
| 6–8 | 7 |
| 8–11 | 3 |
| 11–15 | 2 |
| 15–21 | 0 |

 

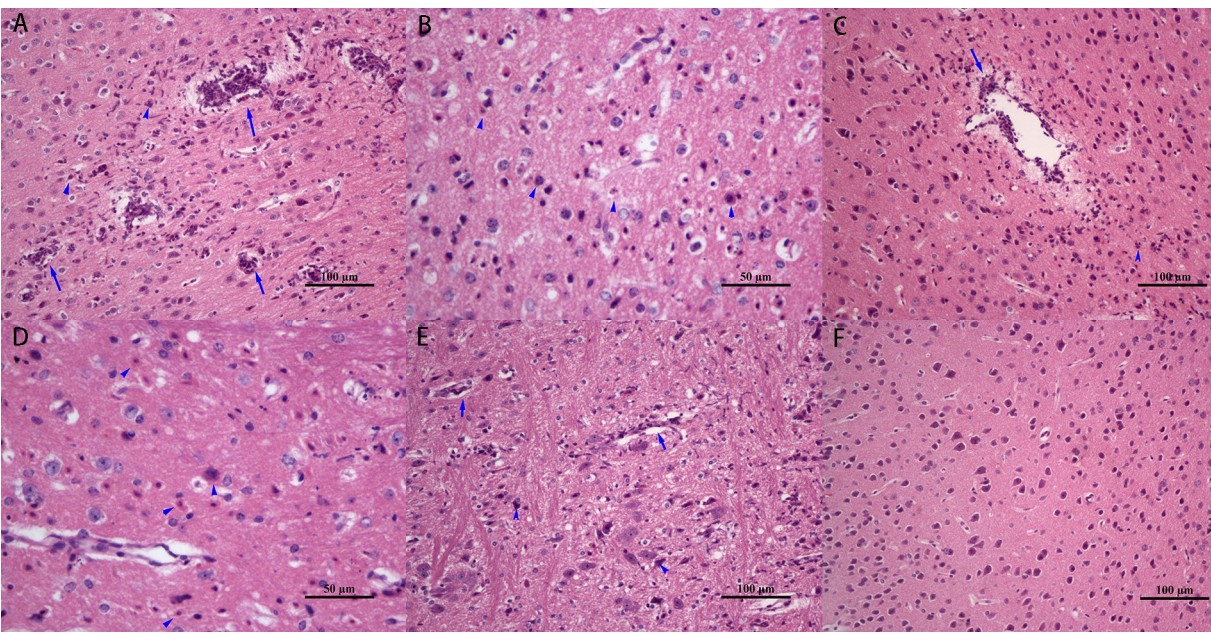

**Fig 1. Dynamic histopathological changes in the brain tissue of JEV infected mice.** Hematoxylin-Eosin staining of the (A) cerebral cortex, 4 dpi, with a large number of lymphocyte and neutrophil infiltration around blood vessels (arrows). (B) cerebral cortex, 6 dpi, with extensive glial cell proliferation, and neuronal degeneration and necrosis (arrowheads). (C) cerebral cortex, 5 dpi, showing neuronal degeneration and necrosis (arrowhead), and perivascular cuffing (arrow). (D) mesencephalon, 6 dpi, showing glial cell proliferation, and neuronal degeneration and necrosis (arrowheads). (E) medulla oblongata, 5 dpi, showing glial cell proliferation and neurophagy (arrowheads). (F) cerebral cortex, control group.

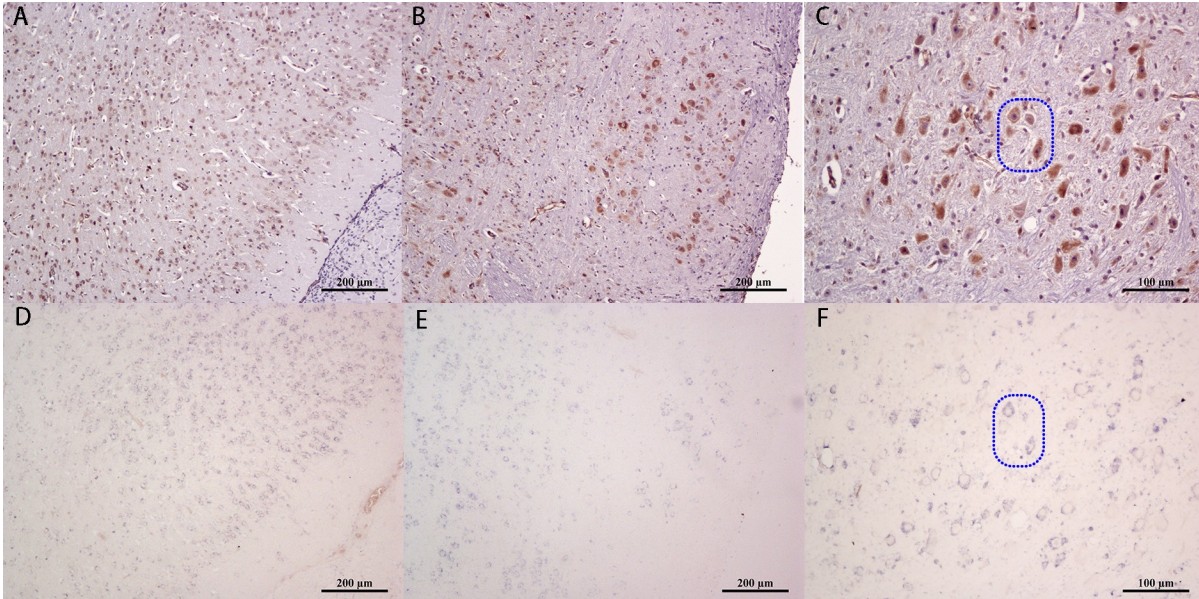

**Fig 2. Localization of JEV in the mouse brain 5 dpi.** (A-C) Immunohistochemical staining of the JEV E protein antigen in the cerebral cortex, medulla oblongata and thalamus, respectively. (D-F) in *situ* hybridization staining of the JEV E gene in the three corresponding regions, respectively (Dotted box).

thalamus (Fig 2C) 5 d after infection. At the same time, ISH showed positive signals for viral nucleic acids distributed in the cytoplasm of corresponding neurons, consistent with the results of IHC (Fig 2D and 2E and 2F, respectively).

Three mice were randomly selected from the left and right brains for virus localization, and no significant differences were found (S1 Fig).

## Validation of nerve nuclei

To determine the accuracy of JEV localization in the nerve nuclei, some nuclei were validated according to their location in the atlas [11]. Tyrosine hydroxylase localization in the brain tissues at 4 dpi was determined by IHC. Positive regions of tyrosine hydroxylase staining were identified in the SN (Fig 3B) and striatum (Fig 3E). Simultaneously, hematoxylin-eosin (Fig 3A and 3D) and IHC staining of JEV were performed on slides of the same tissue, which confirmed the presence of the JEV antigen in the SN (Fig 3C) and striatum (Fig 3F) of mice post virus infection.

## Dynamic localization of viruses in the major cranial nerve nuclei after JEV infection in mice

The IHC results showed that JEV was located in the cytoplasm of the nerve cells. The positive rates of viral antigen in the nerve nuclei at different time points were analyzed. The rate in the reticular thalamic nucleus (Rt) was found to be relatively stable (5–10%) throughout the course of infection and, therefore, the value of each time point was compared with the value of Rt at 4 d.

Viral antigens were first detected 3 dpi, located in the VM, ventral posteromedial thalamic nucleus (VPM), Rt of the diencephalon, the lateral paragigantocellular nucleus (LPGi), and lateral reticular nucleus (LRt) of the medulla oblongata (S2 Fig). At 3.5 dpi, the distribution area

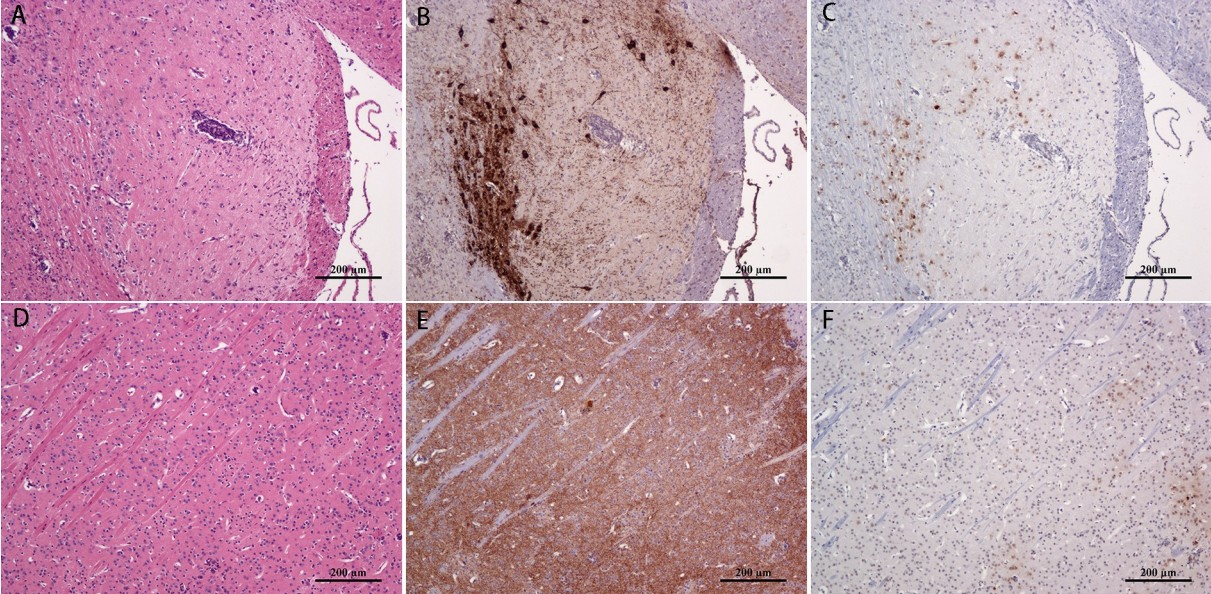

**Fig 3. Co-localization of viral antigens in mouse brain nuclei 4 d post JEV infection.** HE staining of the (A) substantia nigra and (D) striatum. immunohistochemical staining of tyrosine hydroxylase (brown) in corresponding areas of the (B) substantia nigra and (E) striatum. immunohistochemical staining of the JEV E protein antigen (brown) in corresponding areas of the (C) substantia nigra and (F) striatum.

of the virus antigen had slightly increased, but the signal was relatively weaker and the positive rate was less than 5% (S3 Fig).

At 4 dpi, both the distribution and signal intensity of the virus had increased in the brain, and positive rate of the antigen in most nerve nuclei was greater than 5%. Among them, the anterior olfactory nucleus, posterior part (AOP) and anterior olfactory nucleus, ventral part (AOV) in the cerebrum, and VM and VPM antigen in the mesencephalon and brainstem had positive rates above 10%, with a significant difference from that of Rt (P < 0.05, Fig 4A). The positive rates of most viral antigens reached 10% at 5 dpi. The antigens of the AOP, anterior olfactory nucleus, lateral part (AOL) and dorsal part (AOD) in the cerebrum, and posterior thalamic nuclear group (Po), ventrolateral thalamic nucleus (VL), VM, ventral posterolateral thalamic nucleus (VPL), DpMe, Substantia nigra, compact part (SNC) and Mo5 in the mesencephalon and brainstem had positive rates above 15%, with a significant difference relative to Rt (P < 0.01, Fig 4B).

At 6 dpi, the viruses were most widely distributed in the brain (Fig 5A, 5I and 5M). At this time point, a large number of positive signals still existed in the mediodorsal thalamic nucleus, medial part (MDM), central medial thalamic nucleus (CM), mediodorsal thalamic nucleus, lateral part (MDL), centrolateral thalamic nucleus (CL), paracentral thalamic nucleus (PC), Po, VL, VM, VPM (Fig 5E), VPL, and Rt of the diencephalon (Fig 5P). The cingulate cortex, area 1 (Cg1), prelimbic cortex (PrL), dorsal tenia tecta (DTT), ventral tenia tecta (VTT), medial orbital cortex (MO), ventral orbital cortex (VO), lateral orbital cortex (LO), frontal association cortex (FrA), lateral septal nucleus, dorsal part (LSD), lateral septal nucleus, intermediate part (LSI), accumbens nucleus, shell (AcbSh), accumbens nucleus, core (AcbC), anterior olfactory nucleus, medial part (AOM), AOV, AOP, AOL, AOD and anterior olfactory nucleus, external part (AOE) (Fig 5B and 5J), olfactory tubercle (Tu), retrosplenial agranular cortex (RSA), retrosplenial granular b cortex (RSGb), primary motor cortex (M1) (Fig 5L), secondary motor cortex (M2) (Fig 5D), primary somatosensory cortex, hindlimb region (S1HL), primary somatosensory cortex, forelimb region (S1FL), primary somatosensory cortex, trunk region (S1Tr), secondary visual cortex, mediomedial area (V2MM), primary visual cortex (V1), primary somatosensory cortex (S1), caudate putamen (CPu), (Fig 5K), lateral globus pallidus (LGP), piriform cortex (Pir), dorsal endopiriform nucleus (DEn), anterior amygdaloid area, dorsal part (AAD), medial amygdaloid nucleus, anterior dorsal (MeAD), posterolateral cortical amygdaloid nucleus (PLCo), anterior cortical amygdaloid nucleus (ACo) and central amygdaloid nucleus, medial division (CeM) of the cerebrum displayed a high concentration of JEV antigens. The positive signals were densely distributed in the medial mammillary nucleus, lateral part (ML), field CA1 of hippocampus (CA1), field CA2 of hippocampus (CA2), pyramidal cell layer of the hippocampus (Py) (Fig 5C), gigantocellular reticular nucleus (Gi) (Fig 5G), dorsal paragigantocellular nucleus (DPGi), paramedian reticular nucleus (PMn), medullary reticular nucleus, ventral part (MdV) (Fig 5H), medullary reticular nucleus, dorsal part (MdD), parvicellular reticular nucleus (PCRt) and intermediate reticular nucleus (IRt) of the medulla oblongata and Paramedian raphe nucleus (PMnR), Pontine reticular nucleus, oral part (PnO), Pontine reticular nucleus, caudal part (PnC) (Fig 5F), and Mo5 (Fig 5N) of the pons. Positive signals in the SNC and Substantia nigra, reticular part (SNR) (Fig 5O) of the mesencephalon were densely distributed, while those in the superior colliculus (SC), dorsomedial periaqueductal gray (DMPAG), dorsolateral periaqueductal gray (DLPAG) and DpMe were dispersed.

The three-dimensional (3D) images at 6 dpi (Fig 6) showed that the positive signals of viral antigen were concentrated in the parietal lobe, occipital lobe, and hippocampus of the cerebral cortex, which further confirmed the location of the described viral antigens in the nerve nuclei. The positive rates of viral antigens in each nucleus is shown in Fig 4. The positive rate of most viral antigens reached 10%. The antigen positive rates of the AOP, AOV, CA2, and ACo in the

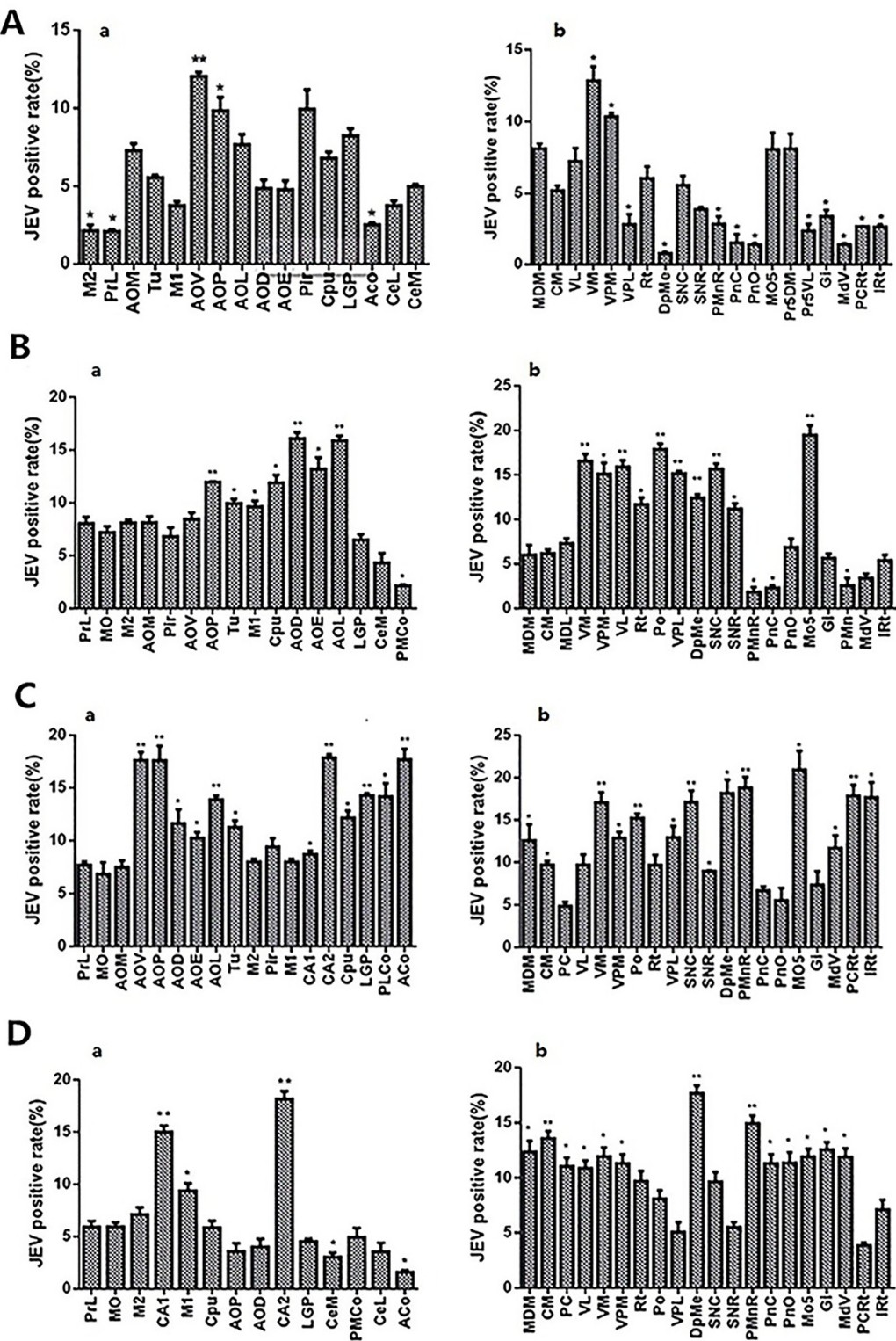

**Fig 4. Statistical changes in the expression of virus in major cranial nerve nuclei at different times in JEV infected mice.** (A-D) represents 4 dpi, 5 dpi, 6 dpi, and 8 dpi, We calculated the positive rate of JEV in the nerve nucleus in each period after infection, and compared it with the positive rate of JEV in Rt on the 3 rd day after infection to show the degree of JEV infection in different periods. (a) the positive rate of JEV in the major cranial nerve nuclei, (b) the positive rate of JEV in the nuclei of diencephalon and brainstem. * P < 0.05, ** P < 0.01, n = 3.

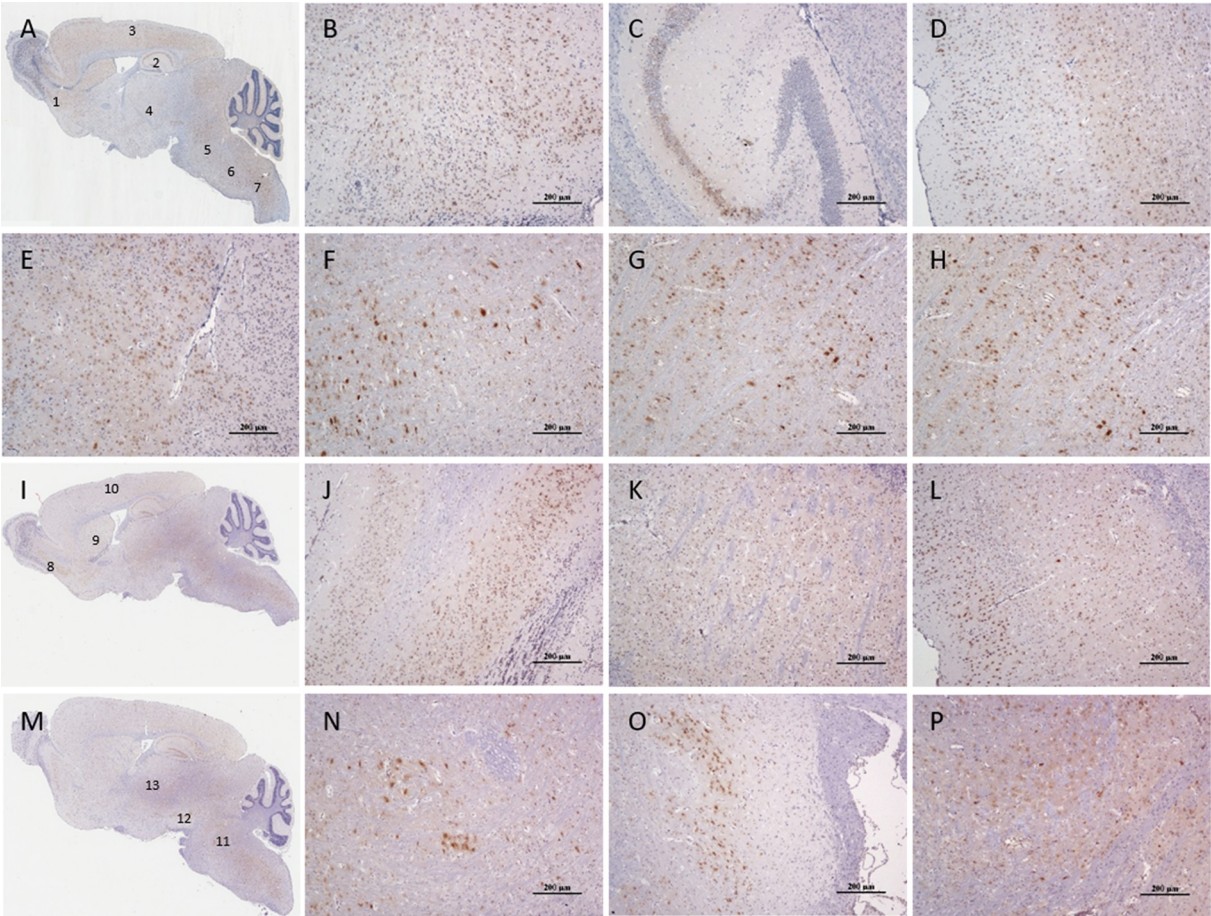

**Fig 5. Localization of virus in mouse cranial nerve nuclei at 6 d post JEV infection.** (A, I, M) The distance of the parallel brain sections from the mid-sagittal plane were 0.72 mm, 0.96 mm, and 1.32 mm, respectively. The enlarged images at 1–13 sites correspond to immunohistochemical staining of the JEV antigen in (B) AOV, AOP, and AOM; (C) CA3, CA2, and Py; (D) M2; (E) VL, VM, and VPM; (F) PnO and PnC; (G) Gi; (H) MdV; (J) AOV, AOP, and AOD; (K) CPu; (L) M1; (N) Mo5; (O) SNC and SNR; (P) Rt and VPL.

cerebrum, and the VM, SNC, and PMnR in the mesencephalon, and brainstem were close to 20%, with a significant difference relative to Rt (P < 0.01, Fig 4C).

From 8 dpi to 15 dpi, the positive signal of JEV in the brain gradually decreased, with the JEV distribution concentrating mainly in the cerebrum and thalamic nuclei. No virus antigen was detected at 21 dpi. The positive rate of viral antigen dropped to 5% in most of the mouse cerebrum at 8 dpi, whereas it remained at 10% in most of the mesencephalon and brainstem. The antigen positive rates of the AOP, AOV, AOL, CA2, and ACo in the cerebrum, and the Po, VM, DpMe, SNC, and Mo5 in the mesencephalon and brainstem were above 15%, which was significantly different than in Rt (P < 0.01, Fig 4D). At 11 dpi, the positive signals in all nuclei except the PrL, CA2, MDM, CM, and PC had decreased to about 5% (S4 Fig) and, after 15 d, the number of positive nuclei decreased further, such that, the positive rate of antigen decreased to approximately 3% (S5 Fig).

In order to visually understand the dynamic distribution of virus in each nucleus, we mapped the pattern of virus distribution at different time points after infection (S6–S13 Figs) based on the mouse anatomical brain atlas [11] and the JEV antigen distribution by IHC staining (S14–S25 Figs). This part used a large number of cranial nerve nuclei, in order to better facilitate readers to read, the abbreviations of the various nuclei are summarized (S1 Table).

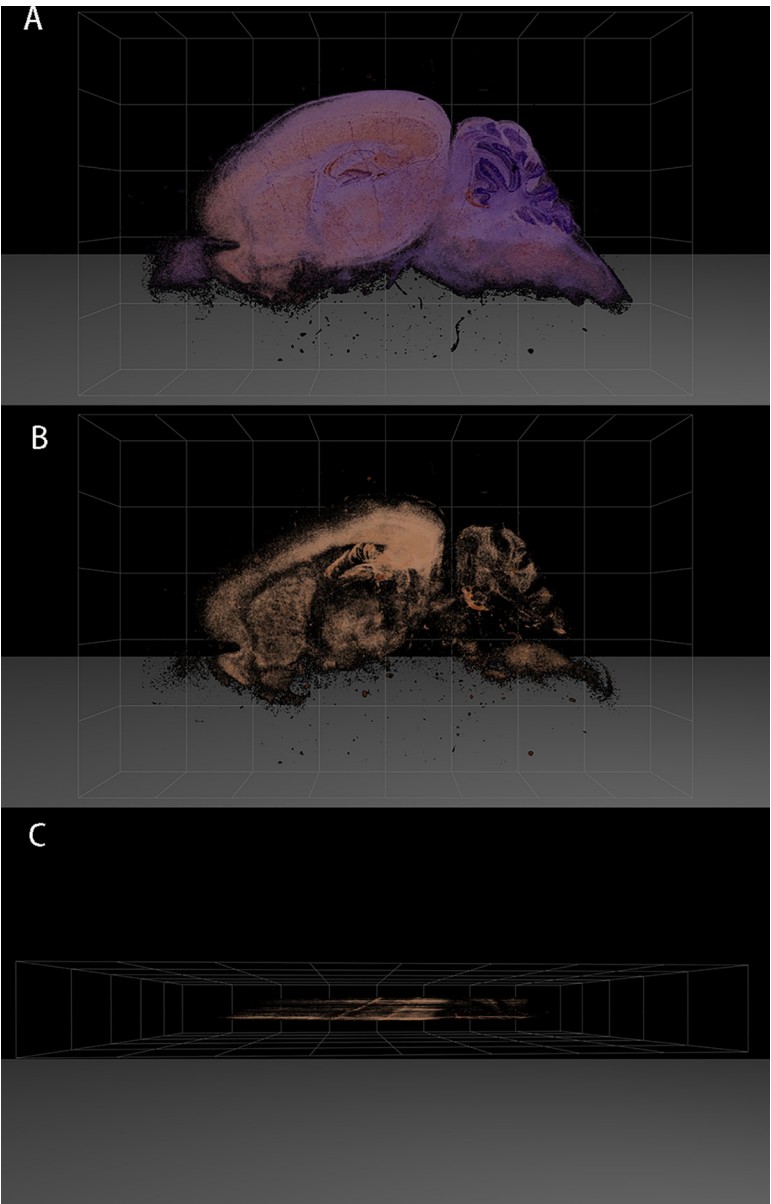

**Fig 6. 3D localization of JEV in the mouse brain at 6 d post JEV infection.** (A) 3D frontal image, from the mid-sagittal side of the brain outward, with the cerebrum on the left and the cerebellum on the right. (B) 3D frontal image with background removed, showing immunohistochemical staining of the virus antigen (brown). (C) 3D overhead view with the background removed, showing staining of the virus antigen (brown).

## Discussion

In this study, the JEV infected mice displayed typical neurological signs. The pathological changes in the brain tissue were widely distributed in the cerebral cortex, basal ganglia, thalamus, mesencephalon, pons, and medulla oblongata. They had the characteristics of non-suppurative encephalitis, including degeneration and necrosis of neurons, glial cell proliferation, and perivascular cuffing. The results showed that JEV was neurotropic and caused noticeable damage to the brain tissue, consistent with results from previous studies [10,12]. In addition, we also focused on the dynamic changes in histopathology. At 4 dpi, mainly inflammation

around blood vessels and proliferation of microglia were observed. From 5 dpi to 6 dpi, extensive neuronal necrosis occurred, and inflammatory cell infiltration around blood vessels were visible. At 8 dpi, perivascular cuffing was rarely observed, although notable glial cell proliferation and neuronal necrosis were detected. These results are similar to pathological changes reported in human autopsy studies [9].

Entry of JEV into the CNS may occur in a variety of ways, either directly or through endothelial cells [13], or through the intact blood-brain barrier [14]. Another possible mechanism is the Trojan horse method, in which the virus uses infected white blood cells to cross the blood-brain barrier into the CNS. It has been reported that certain pathogens can enter and travel in the peripheral axons, and eventually reach the neuronal cell bodies in the CNS [15]. Studies have shown that many viruses, including JEV, can enter the CNS through the olfactory pathway [16,17]. In the present study, the earliest detection of viral antigens in the brain tissue was at 3 dpi, when the virus was localized in the nuclei of the thalamus and medulla, with more viral signal subsequently detected in the AO nucleus. It is speculated that this may be related to the entry of the virus into the brain through blood-derived or peripheral nerve pathways [15]. However, the mechanisms of transport through the olfactory nerve, and the subsequent diffusion into the CNS, are still unclear. After the virus invades the neurons, it proliferates rapidly. The virus was found to be extensively distributed throughout the CNS at 4 dpi, suggesting that it may have broken through the blood-brain barrier at 3 dpi and proliferated in the CNS unregulated, resulting in irreversible damage. However, the mechanism by which the virus passes through the blood-brain barrier is not clear. Whether the blood-brain barrier opens after the virus passes through, or whether the virus enters after the blood-brain barrier has opened, remains to be confirmed.

At 6 dpi, the positive signal of JEV in the brain was at its highest, and the distribution at its most extensive. At 21 dpi, no viral antigen was detected in the brain tissue. It is speculated that the reduction of positive signal of JEV may be related to the neutralization of antibody production and increased cellular immunity in the CNS [18]. In addition, cytotoxic T lymphocytes may play an important role in the clearance and control of JEV [19]. In this study, we observed that dyskinesia was more significant in mice 6 dpi than at 3.5 dpi. There was no significant improvement over time up to 8 dpi, although the amount of virus had decreased significantly by this time point. It is possible that JEV caused serious, irreversible damage to neurons, and it has been shown that JEV activates monocytes, macrophages, and glial cells, and induces severe inflammatory responses [14,20,21].

The nerve nuclei with strong antigenic affinity for JEV include the early AO nucleus and the late hippocampal CA2 region. Both the AO and hippocampal CA2 have anatomical and odor-information transmission connections [22]. Early olfactory dysfunction in Parkinson's patients and Lewy corpuscles, a typical lesion in the brain, first appeared in the neurons of the AO nucleus [23]. Studies have shown that West Nile virus can be transported through axons [24]. However, it is not known whether JEV moves between the neuronal bodies of the AO and CA2 nuclei. The positive signal of JEV in the VM, DpMe, and Mo5 were high throughout the experiment. The VM is an important nucleus in the thalamus, which participates in the formation of the cortex-basal ganglia-thalamus-cortex motor loop [25,26] and in the regulation of awakening behavior [27]. At the same time, DpMe neurons also play an important role in wakefulness and sleep [28]. Abnormal motor activity may lead to brain excitation or inhibition, which may reflect the excitatory or lethargic behavior clinically observed in mice. Histopathological analysis of the motor unit of Mo5 showed that an abnormal neuronal structure in this area led to chewing problems in mice [29]. Therefore, we speculate that neuron damage in this area may be related to the clinical signs of teeth grinding in mice.

The clinical manifestations of viral encephalitis are determined by the affinity and persistence of viruses in different brain regions. The clinical signs associated with the nervous system

in JEV patients include obvious dysfunction of consciousness, cognitive, and motor function, as well as central respiratory failure [2]. In this study, JEV antigens were detected in the reticular structures of the MdD, MdV, PMn, LRt, Gi, PCRt, IRt, PnO, and PnC, and thalamus, with varying extent of damage. This may be due to the infection and destruction of reticular junction structures and other neurons in the brainstem, along with the thalamic neurons, resulting in deep coma and respiratory failure [30,31]. Studies have reported a correlation between neurotransmitter levels and behavioral changes in CNS diseases [32,33]. Damage to the SN and caudate putamen (CPu) leads to a decrease in dopamine synthesis, and a decrease in the activity of serotoninergic neurons may also lead to motor disorders [34]. In the present study we found that, after JEV infection, mice showed signs such as back arching, tremor, and hind limb paralysis, while virus antigens were detected in the SN, CPu, and PMMR. Damage to these nuclei caused by the virus may result in a decrease in dopamine and serotonin, leading to motor dysfunction in mice. The above description only partially explains the relationship between JEV infected nerve nuclei and the clinical signs. Further study is warranted to elucidate the relationship between JEV induced injury and clinical signs following the proliferation of neurons in other nerve nuclei.

## Materials and methods

### Ethics statement

This study was conducted within the Guidelines of Regulations for the Administration of Laboratory Animals (Reversion of the State Science and Technology Commission of the People's Republic of China on March 1, 2017). All animals used in this study received prior approval from the Hubei Provincial Experimental Animal Manage Committee and Huazhong Agricultural University Academic Committee. Hubei Provincial Laboratory Animal Quality Certificate Number was HZAUMO- 2017–044, approved by The Scientific Ethic Committee of Huazhong Agricultural University.

### Virus strain and preparation

The JEV P3 strain (GenBank accession no. U47032.1) was provided by the State Key Laboratory of Agricultural Microbiology, Huazhong Agricultural University, Wuhan, China. The virus was propagated in the brains of lactating mice, and the titer of the virus was determined by plaque assay in BHK-21 cells [35].

### Animal infection experiment

Six-week-old female BALB/c mice were used in this study and the Regulations on the Administration of Laboratory Animals in Hubei Province were strictly followed. The 133 mice were divided into two groups, 120 in the infection group and 13 in the control group. A total of 120 mice were intraperitoneally injected with $10^6$ PFU of JEV strain P3, while the control group was injected with the same volume of sterile DMEM. Tissue sampling was conducted at 1, 1.5, 2, 2.5, 3, 3.5, 4, 5, 6, 8, 11, 15 and 21 day post infection, with 4 mice at each time point. The mice were fixed by transcardial perfusion with 4% formaldehyde, and whole brains were harvested. Throughout the experimental period, the clinical signs of the mice were observed regularly, and mortality was recorded.

### Preparation of paraffin sections

After 2 d of fixation with 4% formaldehyde, the brains were removed from the fixation solution and cut along the median sagittal plane, dehydrated, then embedded in paraffin with the

cut side facing downward. Sets of 3 consecutive slices (5 μm thick) were collected at 0.1 mm intervals for the entire length of the tissue. For each set of 3 slices, 1 section was used for HE staining, for analysis of pathological changes in the brain under the microscope, while the other two slides were used for immunohistochemistry and *in situ* hybridization, to locate the virus.

## Immunohistochemistry

Paraffin sections were placed in 3% $H_2O_2$ for 30 min to quench endogenous peroxidase activity. The sectioned slides were incubated in citrate buffer at 96˚C for 30 min, for antigen retrieval. After washing, the sections were blocked in 5% bovine serum albumin for 1 h and incubated overnight at 4˚C with mouse anti-JEV primary antibody (1:200, Mouse monoclonal antibodies against JEV E proteins were provided by the State Key Laboratory of Agricultural Microorganisms. The purified E protein was injected into the mouse, and the spleen cells were fused with the murine myeloma cells. The positive hybridoma cells were selected for intraperitoneal injection into the mouse and ascites was collected. The IgG was purified by commercial kit. This method is described in Monoclonal antibodies against NS3 and NS5 proteins of Japanese encephalitis virus [36] or rabbit anti-tyrosine hydroxylase primary antibody (1:500, Wuhan Servicebio Technology Co., Ltd, China). After washing, the sections were incubated in secondary antibody for 45 min (sheep anti-mouse/rabbit IgG labeled with horseradish peroxidase, Beijing ZSGB-BIO Co., Ltd.). Finally, the slides were developed using DAB, and hematoxylin was used for counterstaining. All immunohistochemical sections were scanned with a Leica Apero CS2 slide scanning system. The distribution of the virus in the brain was analyzed with reference to the relevant literature [11].

## *In situ* hybridization

A specific primer pair (F: 5'-TGGGACTTTGCTATTGG-3'; R: 5'-AGAACACGAGCACGCC TCCT-3') was designed according to the conserved gene sequence of the JEV E protein. The JEV E gene was amplified by PCR (208 bp). After the product was obtained and verified, the sequence was labeled with digoxigenin using the DIG High Prime DNA Labeling and Detection Starter Kit I (Roche Diagnostics Deutschland GmbH, Germany). After dewaxing in water, the sections were pretreated with 0.2 mol/L hydrochloric acid and proteinase K. The probe and tissue were hybridized first at 95˚C for 7 min and then 42˚C overnight. Anti-digoxigenin-AP conjugate was added after blocking, and NBT-BCIP was used for developing for 1 h at 37˚C.

## Three-dimensional distribution of JEV in the mouse brain at 6 d post JEV infection

Consecutive sectioning was performed parallel to the mid-sagittal plane and toward the lateral part of the brain, and serial numbers were assigned to each slide. The thickness of the slides was 5 μm. The odd number of tissue sections were used for virus antigen localization by immunohistochemical staining. The stained tissue sections were scanned by Leica Aperio CS2 digital pathological scanner, and the 3D image of JEV distribution in the brain was constructed using 3D HISTECH & microDimensions software, version 3D View v2.2.0.

## Statistical analysis

The scanned digital sections were analyzed using Aperio ImageScope software, and the positive rate of JEV in the major nerve nuclei was measured according to the atlas location [11] (3–

5 images per nucleus were counted to calculate the percentage of the optical density of the virus-positive signal, and the nerve nucleus area). Graph Pad Prism V5.0 software was used to analyze the differences and to plot the graph. All data were compared with the positive rate of JEV in the reticular thalamic nucleus at 4 dpi.

## Supporting information

**S1 Fig. Comparison of viral antigen signal distribution in the same parts of left and right brain of mice.** The same positive signals area appear in the three typical parts. Brain hippocampus A (left), B (right); cerebral cortex C (left), D (right) and medulla oblongata E (left), F (right). **(IHC)**
(TIF)

**S2 Fig. The distribution of virus antigens in the cerebral nucleus of the mice at 3 d post JEV infection.** $^*$ P $<$ 0.05, $^{**}$ P $<$ 0.01, n = 3.
(TIF)

**S3 Fig. The distribution of virus antigens in the cerebral nucleus of mice at 3.5 d post JEV infection.** $^*$ P $<$ 0.05, $^{**}$ P $<$ 0.01, n = 3.
(TIF)

**S4 Fig. The distribution of virus antigens in the cerebral nucleus of mice at 11 d post JEV infection.** $^*$ P $<$ 0.05, $^{**}$ P $<$ 0.01, n = 3.
(TIF)

**S5 Fig. The distribution of virus antigens in the cerebral nucleus of mice at 15 d post JEV infection.** $^*$ P $<$ 0.05, $^{**}$ P $<$ 0.01, n = 3.
(TIF)

**S6 Fig. Distribution pattern of JEV in each nucleus of the mouse brain at 3d post JEV infection.** Plotted according to the reference literature [11], the red origin point represents the location of JEV distribution. Plotted according to the reference literature [11], the red origin point represents the location of JEV distribution. (A-D) Sagittal images of the brain at a distance of 0.96, 1.08, 1.20, and 1.44 mm from the mid-sagittal plane, respectively.
(TIF)

**S7 Fig. Distribution pattern of JEV in each nucleus of the mouse brain at 3.5d post JEV infection.** Plotted according to the reference literature [11], the red origin point represents the location of JEV distribution. (A-L) Sagittal images of the brain at a distance of 0.12, 0.24, 0.48, 0.60, 0.72, 0.84, 0.96, 1.08, 1.20, 1.32, 1.92, and 2.04 mm from the mid-sagittal plane, respectively.
(TIF)

**S8 Fig. Distribution pattern of JEV in each nucleus of the mouse brain at 4d post JEV infection.** Plotted according to the reference literature [11], the red origin point represents the location of JEV distribution. (A-L) Sagittal images of the brain at a distance of 0.24, 0.36, 0.72, 0.96, 1.20, 1.32, 1.80, 2.04, and 2.16 mm from the mid-sagittal plane, respectively.
(TIF)

**S9 Fig. Distribution pattern of JEV in each nucleus of the mouse brain at 5d post JEV infection.** Plotted according to the reference literature [11], the red origin point represents the location of JEV distribution. (A-L) Sagittal images of the brain at a distance of 0.12, 0.24, 0.60, 0.72, 0.96, 1.08, 1.32, 1.44, 1.68, 1.92, 2.28, and 2.40 mm from the mid-sagittal plane,

respectively.
(TIF)

**S10 Fig. Distribution pattern of JEV in each nucleus of the mouse brain at 6d post JEV infection.** Plotted according to the reference literature [11], the red origin point represents the location of JEV distribution. (A-L) Sagittal images of the brain at a distance of 0.12, 0.24, 0.60, 0.72, 0.96, 1.08, 1.32, 1.44, 1.80, 1.92, 2.28, and 2.52 mm from the mid-sagittal plane, respectively.
(TIF)

**S11 Fig. Distribution pattern of JEV in each nucleus of the mouse brain at 8d post JEV infection.** Plotted according to the reference literature [11], the red origin point represents the location of JEV distribution. (A-L) Sagittal images of the brain at a distance of 0.24, 0.72, 0.84, 0.96, 1.08, 1.68, 1.92, 2.16, 2.52, 2.64, 2.88, and 3.12 mm from the mid-sagittal plane, respectively.
(TIF)

**S12 Fig. Distribution pattern of JEV in each nucleus of the mouse brain at 11d post JEV infection.** Plotted according to the reference literature [11], the red origin point represents the location of JEV distribution. (A-L) Sagittal images of the brain at a distance of 0.12, 0.24, 0.60, 0.72, 0.96, 1.08, 1.20, 1.80, 1.92, 2.04, 2.16, and 2.52 mm from the mid-sagittal plane, respectively.
(TIF)

**S13 Fig. Distribution pattern of JEV in each nucleus of the mouse brain at 15d post JEV infection.** Plotted according to the reference literature [11], the red origin point represents the location of JEV distribution. (A-F) Sagittal images of the brain at a distance of 0.24, 0.60, 0.96, 1.32, 2.04, and 2.64 mm from the mid-sagittal plane, respectively.
(TIF)

**S14 Fig. The distribution of JEV antigen in the brain of mice at 3 d post-infection(medulla oblongata, diencephalon).** (A) VM. (B) LPGi, LRt. **(IHC)**.
(GIF)

**S15 Fig. The distribution of JEV antigen in the brain of mice at 4 d post-infection(diencephalon, cerebrum).** (A) MDM, CM. (B) VL, VM, VPM. (C) VPL, Rt. (D) MO, PrL. (E) DTT, VTT. (F) M1 (G) M2. (H) Tu. (I) AOM, AOV, AOP. (J) AOD, AOL AOV, AOP. (K) CPu. (L) CeL, CeM. **(IHC)**.
(GIF)

**S16 Fig. The distribution of JEV antigen in the brain of mice at 4 d post-infection(medulla oblongata, pons, midbrain).** (A) Gi. (B) PMn. (C) PCRt, IRt. (D) MdD. (E) PMnR. (F) PnO, PnC. (G) Mo5. (H) Pr5VL, Pr5DM. (I) SNC, SNR. **(IHC)**.
(GIF)

**S17 Fig. The distribution of JEV antigen in the brain of mice at 5 d post-infection(diencephalon, cerebrum).** (A) MDM, CM. (B) IAD, AM. (C) VM, VL, PC. (D) CL, MDL. (E) VA, VL. (F) Rt, VPL. (G) PrL, MO. (H) DTT, VTT. (I) AcbC, AcbSh. (J) M1. (K) M2. (L) AOD, AOV, AOP. (M) Tu. (N) Pir, DEn. (O) CPu. (P) CeM, BMP.**(IHC)**.
(GIF)

**S18 Fig. The distribution of JEV antigen in the brain of mice at 5 d post-infection(medulla oblongata, pons, midbrain).** (A) Gi. (B) PMn. (C) PCRt, IRt. (D) MdD, MdV. (E) PMnR. (F)

PnO, PnC. (G) VLTg, Pn. (H) Mo5. (I) SuG, SC. (J) DMPAG. (K) DpMe. (L) SNR, SNC. **(IHC)**.
(GIF)

**S19 Fig. The distribution of JEV antigen in the brain of mice at 6 d post-infection(diencephalon, cerebrum).** (A) MDM, CM. (B) VL, VM, VPM. (C) Po, VL. (D) Rt, VPL. (E) Cg1. (F) PrL, MO. (G) LSD. (H) CA1. (I) CA2, CA3, Py. (J) M2. (K) M1. (L) LPO. (M) ML. (N) CPu. (O) AOV, AOP, AOD. (P) AcbC, AcbSh. (Q) Pir, DEn. (R) AAD, MeAD. (S) ACo, PLCo (T) CeM. **(IHC)**.
(GIF)

**S20 Fig. The distribution of JEV antigen in the brain of mice at 6 d post-infection(medulla oblongata, pons, midbrain).** (A) Gi. (B) MdV. (C) PCRt, IRt. (D) PMnR. (E) PnO, PnC. (F) Mo5. (G) DMPAG. (H) DpMe. (I) SNC, SNR. **(IHC)**.
(GIF)

**S21 Fig. The distribution of JEV antigen in the brain of mice at 8 d post-infection(diencephalon, cerebrum).** (A) MDM, CM. (B) CL, MDL. (C) VL, VM, VPM. (D) PrL, MO. (E) LSD, LSI. (F) MM, ML. (G) CA1. (H) CA2. (I) M2. (J) M1. (K) AOP. (L) AOP. (M) CPu. (N) MePD, MePV. (O) ACo. (P) CeL, CeM. **(IHC)**.
(GIF)

**S22 Fig. The distribution of JEV antigen in the brain of mice at 8 d post-infection(medulla oblongata, pons, midbrain).** (A) Gi, DPGi. (B) PMn. (C) PMnR. (D) Pn, RtTg. (E) PnO, PnC. (F) Mo5. (G) InWh, DpWh. (H) DpMe. (I) SNC, SNR. **(IHC)**.
(GIF)

**S23 Fig. The distribution of JEV antigen in the brain of mice at 11 d post-infection(diencephalon, cerebrum).** (A) MDM, CM. (B) MDL, PC. (C) VL, VM, VPM. (D) Rt, VPL. (E) PrL, MO. (F) CA1. (G) CA2. (H) M2. (I) M1. (J) MM, ML. (K) CPu. (L) AOM, AOV, AOP. (M) Pir. (N) AAV, AAD. (O) ACo, PLCo. (P) CeM. **(IHC)**.
(GIF)

**S24 Fig. The distribution of JEV antigen in the brain of mice at 11 d post-infection (medulla oblongata, pons, midbrain).** (A) Gi, DPGi. (B) PCRt, IRt. (C) Pn, RtTg. (D) PMnR. (E) PnO, PnC. (F) Mo5. (G) SuG, SC. (H) SNC, SNR. **(IHC)**.
(GIF)

**S25 Fig. The distribution of JEV antigen in the brain of mice at 15 d post-infection(diencephalon, cerebrum).** (A) MDM. (B) VPM. (C) PrL. (D) M2. (E) CPu. (F) Tu. (G) Pir, DEn. (H) AAD, AAV. (I) ACo. **(IHC)**.
(GIF)

**S1 Table. Abbreviated form.**
(DOCX)

## Author Contributions

**Conceptualization:** Changqin Gu.

**Data curation:** Mingxing Gao, Changqing Xie.

**Formal analysis:** Jinhua Zhang, Wanpo Zhang.

**Funding acquisition:** Shengbo Cao, Changqin Gu.

**Investigation:** Wei Han, Changqin Gu.

**Methodology:** Zikai Zhao, Xueying Hu, Xiaoli Liu.

**Project administration:** Changqin Gu.

**Resources:** Shengbo Cao, Guofu Cheng.

**Software:** Wei Han.

**Supervision:** Changqin Gu.

**Validation:** Wei Han.

**Visualization:** Changqin Gu.

**Writing – original draft:** Wei Han, Changqin Gu.

**Writing – review & editing:** Jinhua Zhang.

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
