## [Decision Letter · Decision Letter 0]

1 Aug 2020

Dear Dr Changqin,

Thank you very much for submitting your manuscript "Precise Localization and Dynamic Distribution of Japanese Encephalitis Virus in the

Brain Nuclei of Infected Mice" for consideration at PLOS Neglected Tropical Diseases. As with all papers reviewed by the journal, your manuscript was reviewed by members of the editorial board and by several independent reviewers. In light of the reviews (below this email), we would like to invite the resubmission of a significantly-revised version that takes into account the reviewers' comments. 

We cannot make any decision about publication until we have seen the revised manuscript and your response to the reviewers' comments. Your revised manuscript is also likely to be sent to reviewers for further evaluation.

Sincerely,

Michael R Holbrook, PhD

Associate Editor

Sunit Singh

Deputy Editor

Reviewer's Responses to Questions

**Key Review Criteria Required for Acceptance?**

**Methods**

-Are the objectives of the study clearly articulated with a clear testable hypothesis stated?

-Is the study design appropriate to address the stated objectives?

-Is the population clearly described and appropriate for the hypothesis being tested?

-Is the sample size sufficient to ensure adequate power to address the hypothesis being tested?

-Were correct statistical analysis used to support conclusions?

-Are there concerns about ethical or regulatory requirements being met?

Reviewer #1: Meaning of P3 in the abstract isn’t clear. I suggest changing it to “low passage” and also clearly writing this in the methods. Can they provide the source of the virus as animal or human since this is a key point of the paper.

The paper uses a non-commercial antibody for detection of JEV and it is not written how that antibody was generated. The authors just say “anti-JEV” but it should be more specific what the antigen was and the level of purification of that antigen. A single protein? Purified virus? Etc..

Reviewer #2: (No Response)

Reviewer #3: The study was carried out to localize the JEV in the CNS of mice at a different times post infection, which will be helpful to elucidate the pathogenesis and distribution of the virus in the host. The study design is partly addressed the stated objectives. The sample size is sufficient to ensure adequate power to address the hypothesis being tested. Correct statistical analysis were used. There are concers about ethical requirements being met.

**Results**

-Does the analysis presented match the analysis plan?

-Are the results clearly and completely presented?

-Are the figures (Tables, Images) of sufficient quality for clarity?

Reviewer #1: Figure 1d, each of the features, “glial cell proliferation” “neuronal degeneration” and “necrosis” should be indicated with their own symbols or arrowheads, as is it is not clear. For e, arrows also needed. The aarows are quite small and it would be easier to see if inset images are used. Figures 1-2 could be combined. Control, uninfected tissue is needed. 

Fig. 3 legend is out of order, making it difficult to follow. Colocalization cannot be determined from this since it is clearly used on different sections. Staining in similar regions, perhaps, but a technique like confocal is needed to confirm colocalization. 

Figure 4 is also not clear how the calculations are being made. Are they based on images similar to those presented in Figure 3? Acronyms in the Fig. 4 legend make it inaccessible. Figures 5-6 legends don’t seem to match. I don’t see any statistics in Fig 6. The image provided in Fig. 6c is not informative. 

Much of the information in the supplemental materials should be included in in main figures, particularly those parts describing the course of infection, symptoms, and virus levels.

Reviewer #2: (No Response)

Reviewer #3: The analysis presented match the analysis plan. The results are clearly and completely presented. The figures are sufficient quality for clarity

**Conclusions**

-Are the conclusions supported by the data presented?

-Are the limitations of analysis clearly described?

-Do the authors discuss how these data can be helpful to advance our understanding of the topic under study?

-Is public health relevance addressed?

Reviewer #1: It should be discussed that this strain causes a non-lethal infection, unlike the Nakayama strain which is well characterized and produces lethal disease in mice. In general, the authors do not cite enough studies that describe histology/pathology of mouse brains in experimental JEV infection. 

As mentioned elsewhere, the conclusions of colocalization are weak and not supported by techniques that would allow those statements.

Reviewer #2: (No Response)

Reviewer #3: The conclusions are supported by the data presented. The authors discuss that these data explains the relationship between JEV infected nerve nuclei and the clinial symptoms. No public health relevance is addressed.

**Editorial and Data Presentation Modifications?**

Reviewer #1: Much of the supplemental needs to be included as main figures.

Reviewer #2: (No Response)

Reviewer #3: (No Response)

**Summary and General Comments**

Reviewer #1: This manuscript uses a non-lethal strain of JEV and identifies areas of infection in the brain of experimentally infected mice. This is a study to describe the regions of the brain where JEV is detected. An emphasis is placed on the localization of JEV to “nerve” nuclei, but there is insufficient dual-staining with markers to identify the localization of JEV at a cellular level. The manuscript order made it very difficult to review. There also appear to be some major issues with the matching of figure legends to figures that make the manuscript impossible to fully assess in the current format. The quality needs to be improved significantly before this would be ready for in-depth review and publication.

Reviewer #2: This manuscript by Han and colleagues describes a Japanese encephalitis virus (JEV) neuropathology model in mice. The longitudinal study design allowed the investigators to examine JEV antigen distribution and pathology at different times post infection. While nothing particularly novel was discovered in this study that has not been described previously for JEV or other neurotropic flaviviruses, the work appears to be well done, is very detailed, and supports observations described in the literature. Furthermore, the longitudinal brain sampling permits some inferences as to the mode of entry of JEV into the central nervous system (CNS). More specific comments are below.

Major comments:

1. While the intraperitoneal (IP) route of inoculation is an acceptable way to challenge mice with JEV, it does not really mimic natural infection. It is possible that virus replication kinetics post IP inoculation vary greatly from more natural routes of exposure such as intradermal (ID). The route of inoculation could also impact the mode of virus neuroinvasion. This is a major limitation of this mouse model. Did the authors consider using ID inoculation? Furthermore, can the authors comment on the use of a 10^6 PFU challenge dose (line 367 of the manuscript). This dose is quite high and could also impact the factors listed above.

Minor comments:

2. Line 39, what is meant by “consistent concentration”. Was a concentration measured?

3. Throughout the manuscript, hours post infection are reported. Consider also indicating the number of days in parentheses as many researchers are accustomed to seeing time post infection represented this way.

4. Line 82, mice are “a good” model for JEV, I would not consider them to be “an ideal”, as stated here. Please consider revising.

5. Throughout the manuscript, change “symptoms” to “signs” when referring to mouse studies.

6. Line 91, consider changing “fluffy coat” to “ruffled fur”, which is more commonly used. Also, what is meant by “curling behavior”? Are the authors referring to “hunched posture”?

7. The paragraph starting on line 176 contains sentences with very long lists of brain regions and it is very difficult to read. Can a Table/Figure be referenced and a very general statement be made in the text instead of listing everything.

8. Figure 5 and 6 legends do not match with the actual Figures.

9. Line 308, was the “viral load” in the brain measured? Usually this refers to infectious virus or viral genome copy measurements. Consider rewording.

10. Line 396, is the “anti-JEV primary antibody” mouse antiserum? How was it generated? Please indicate in the manuscript.

Reviewer #3: In this article, Wei Han et al established a mouse model infected with Japanese encephalitis virus (JEV). The virus dynamical localization and the distribution were depicted through the IHC and ISH and construction of three-dimensional (3D) images. The results showed that the viral localization and distribution changed in a time-dependent manner and consistent with the clinical symptoms. Although the detailed dynamic localization and distribution will be helpful to know the virus entry pathway and transmission, it seemed very weak and ineffective for clarifying the pathogenesis of the virus. No clear innovation was discovered for the transmissible mechanism of the disease. It may suggest the authors should localization how the virus passes through the blood-brain barrier.

PLOS authors have the option to publish the peer review history of their article (what does this mean?). If published, this will include your full peer review and any attached files.

Reviewer #1: No

Reviewer #2: No

Reviewer #3: No
---

## [Editor Report · Decision Letter 1]

9 Jan 2021

Dear Dr Changqin,

Thank you very much for submitting your manuscript "Precise Localization and Dynamic Distribution of Japanese Encephalitis Virus in the Brain Nuclei of Infected Mice" for consideration at PLOS Neglected Tropical Diseases. 

As with all papers reviewed by the journal, your manuscript was reviewed by members of the editorial board prior to sending out for re-review. While there have been modifications to the original submission, several author comments, particularly those from reviewer 1 were not adequately addressed in the manuscript. Putting comments in a response to the reviewer is insufficient when the comments specifically address critical concerns with the manuscript. One of the authors also suggested moving some of the supplemental figures to the main text. This was neither performed nor adequately addressed in my view. The bar graphs in the supplemental figures could easily be added to the main text and would enhance the submission. In addition, the legends for supplemental figures S6-S13 refer to "images" which implies that these are photomicrographs when they are in fact brain atlases with JEV antigen distribution indicated. Please clarify in the legends.

We would like to invite the resubmission of a significantly-revised version that takes into account the editors' comments above.

We cannot make any decision about publication until we have seen the revised manuscript and your response to the reviewers' comments. Your revised manuscript is also likely to be sent to reviewers for further evaluation.

Sincerely,

Michael R Holbrook, PhD

Associate Editor

Sunit Singh

Deputy Editor
---

## [Decision Letter · Decision Letter 2]

15 Mar 2021

Dear Dr Changqin,

Thank you very much for submitting your manuscript "Precise Localization and Dynamic Distribution of Japanese Encephalitis Virus in the

Brain Nuclei of Infected Mice" for consideration at PLOS Neglected Tropical Diseases. As with all papers reviewed by the journal, your manuscript was reviewed by members of the editorial board and by several independent reviewers. The reviewers appreciated the attention to an important topic. Based on the reviews, we are likely to accept this manuscript for publication, providing that you modify the manuscript according to the review recommendations. 

Sincerely,

Michael R Holbrook, PhD

Associate Editor

Sunit Singh

Deputy Editor

Reviewer's Responses to Questions

**Key Review Criteria Required for Acceptance?**

**Methods**

-Are the objectives of the study clearly articulated with a clear testable hypothesis stated?

-Is the study design appropriate to address the stated objectives?

-Is the population clearly described and appropriate for the hypothesis being tested?

-Is the sample size sufficient to ensure adequate power to address the hypothesis being tested?

-Were correct statistical analysis used to support conclusions?

-Are there concerns about ethical or regulatory requirements being met?

Reviewer #3: (No Response)

Reviewer #4: The methods are clearly explained. However, some edits are needed.

1) Line 372- not sure what the sentence means.

2) Please refrain from describing euthanasia methods and delete lines 373-374. 

3) Is 2 days enough to inactivate JEV? (line 376).

**Results**

-Does the analysis presented match the analysis plan?

-Are the results clearly and completely presented?

-Are the figures (Tables, Images) of sufficient quality for clarity?

Reviewer #3: (No Response)

Reviewer #4: The results are described clearly. However, considerable editing is required.

1) Animals do not exhibit symptoms but rather signs. Please replace "symptoms" with "signs".

2) Please do not state that "animals died (Line 95). Every approved animal protocol has euthanasia criteria. Please replace "died" with "euthanized". 

3)

**Conclusions**

-Are the conclusions supported by the data presented?

-Are the limitations of analysis clearly described?

-Do the authors discuss how these data can be helpful to advance our understanding of the topic under study?

-Is public health relevance addressed?

Reviewer #3: (No Response)

Reviewer #4: The conclusions as well supported and stated clearly.

**Editorial and Data Presentation Modifications?**

Reviewer #3: (No Response)

Reviewer #4: Please hyphenate after "post" throughout the manuscript i.e. post-infection.

Please abbreviate "hours post-infection (hpi)" and use "hpi throughout the manuscript.

Please italicize "in situ" throughout the manuscript. 

Please add white space in figure 1 and 2 to distinguish each picture.

**Summary and General Comments**

Reviewer #3: Although the authors have revised greatly according to Reviewer's suggestions, there are still some problems needed to be addressed as showed in following: 

1. Lines 160-162, 226-227: Missing space around parentheses.

2. Page 2, line 32: “which lasted up to 、8d” should be corrected to“which lasted up to 8d”

3. Page 17, line 362：“……21 day post treatment” should be corrected to “……21 day post infection”.

4. It’s difficult to understand statistical analysis in figures of Fig. 4. Were all groups compared with IRt? It’s suggested that the significant differences among groups should be illustrated in the legend or marked in the figures.

5. The legend of figure F in Fig.2 was missing.

6. The revision in line 39 was not found. Reviewer #2: Line 39, what is meant by “consistent concentration”. Was a concentration measured? And the response 2：Thank for your suggestion，we reversed the sentence to“and the virus content is relatively stable”. Please see line 39.

7. The revision in line 82 was not found. Reviewer #2: Line 82, mice are “a good” model for JEV, I would not consider them to be “an ideal”, as stated here. Please consider revising. And the response 4: Thanks for your advice. We have changed “a good animal model” to “an ideal animal model”. Please see line 82.

8. The revision in line 91 was not found. Reviewer #2: consider changing “fluffy coat” to “ruffled fur”

9. The revision in line 327 was not found. Response 9: Thanks for your reasonable suggestions. We have change“viral load” to “positive signal of JEV”. Please see line 327.

10. The revision in line 417-424 was not found. Response 10： Yes，mouse monoclonal antibodies against JEV E proteins were provided by the State Key Laboratory of Agricultural Microorganisms. We have added the details in the method. Please see line 417-424.

Please correct the above points, or it's difficult to accept as the current status.

Reviewer #4: The manuscript details a study that investigated the CNS infection of JEV in mice inoculated via a peripheral route. The study is important as it details a time-course infection in various parts of the brain.

PLOS authors have the option to publish the peer review history of their article (what does this mean?). If published, this will include your full peer review and any attached files.

Reviewer #3: No

Reviewer #4: No

Figure Files:

Data Requirements:

Reproducibility:

References

---

## [Editor Report · Decision Letter 3]

27 Apr 2021

Dear Dr Changqin,

We are pleased to inform you that your manuscript 'Precise Localization and Dynamic Distribution of Japanese Encephalitis Virus in the

Brain Nuclei of Infected Mice' has been provisionally accepted for publication in PLOS Neglected Tropical Diseases.

Best regards,

Michael R Holbrook, PhD

Associate Editor

Sunit Singh

Deputy Editor

Authors are encouraged to very carefully review this submission when given the opportunity. The PLoS editorial staff does not proof submissions.

---

## [Editor Report · Acceptance letter]

1 Jun 2021

Dear Dr Gu,

We are delighted to inform you that your manuscript, "Precise Localization and Dynamic Distribution of Japanese Encephalitis Virus in the
Brain Nuclei of Infected Mice," has been formally accepted for publication in PLOS Neglected Tropical Diseases.

Best regards,

Shaden Kamhawi

co-Editor-in-Chief

Paul Brindley

co-Editor-in-Chief
